# Evaluation of data processing pipelines on real-world electronic health records data for the purpose of measuring patient similarity

**Maria Pikoula**[1]*, **Constantinos Kallis**[2], **Sephora Madjiheurem**[3], **Jennifer K. Quint**[2], **Mona Bafadhel**[4], **Spiros Denaxas**[1]

1 Institute of Health Informatics, University College London, London, United Kingdom, 2 National Heart and Lung Institute, Imperial College London, London, United Kingdom, 3 Department of Electronic and Electrical Engineering, University College London, London, United Kingdom, 4 School of Immunology and Microbial Sciences, King's College London, London, United Kingdom

* m.pikoula@ucl.ac.uk

## Abstract

### Background

The ever-growing size, breadth, and availability of patient data allows for a wide variety of clinical features to serve as inputs for phenotype discovery using cluster analysis. Data of mixed types in particular are not straightforward to combine into a single feature vector, and techniques used to address this can be biased towards certain data types in ways that are not immediately obvious or intended. In this context, the process of constructing clinically meaningful patient representations from complex datasets has not been systematically evaluated.

### Aims

Our aim was to a) outline and b) implement an analytical framework to evaluate distinct methods of constructing patient representations from routine electronic health record data for the purpose of measuring patient similarity. We applied the analysis on a patient cohort diagnosed with chronic obstructive pulmonary disease.

### Methods

Using data from the CALIBER data resource, we extracted clinically relevant features for a cohort of patients diagnosed with chronic obstructive pulmonary disease. We used four different data processing pipelines to construct lower dimensional patient representations from which we calculated patient similarity scores. We described the resulting representations, ranked the influence of each individual feature on patient similarity and evaluated the effect of different pipelines on clustering outcomes. Experts evaluated the resulting representations by rating the clinical relevance of similar patient suggestions with regard to a reference patient.

**Data Availability Statement:** Due to privacy laws and the data user agreement between the University College London and Clinical Practice Research Datalink, authors are not authorised to

share individual patient data from these electronic health records. We are unfortunately unable to share the raw data pertaining to the patient records used, and access is only possible for approved researchers, following CPRD's Research Data Governance Process, as described in the following guide: https://cprd.com/safeguarding-patient-data; this is in line with the data providers governance framework we operate under and while data are indeed anonymized, the risk of identifiability still remains. Requests to access data provided by Clinical Practice Research Datalink (CPRD) should be sent to the Independent Scientific Advisory Committee (ISAC).

**Funding:** This study was supported by Health Data Research UK (award ref.: LOND1), which is funded by the UK Medical Research Council, Engineering and Physical Sciences Research Council, Economic and Social Research Council, Department of Health and Social Care (England), Chief Scientist Office of the Scottish Government Health and Social Care Directorates, Health and Social Care Research and Development Division (Welsh Government), Public Health Agency (Northern Ireland), British Heart Foundation and Wellcome Trust, and the Asthma and Lung UK (award ref.: JRFG18–1). The funding bodies played no role in the design, analysis and interpretation of the data nor in the writing of the manuscript. This study is based in part on data from the Clinical Practice Research Datalink obtained under licence from the UK Medicines and Healthcare products Regulatory Agency. The data is provided by patients and collected by the NHS as part of their care and support. The interpretation and conclusions contained in this study are those of the author/s alone.

**Competing interests:** The authors declare the following financial interests/personal relationships which may be considered as potential competing interests: Mona Bafadhel has received grants from AstraZeneca, Roche (to institution). Has received honoraria from AstraZeneca, Chiesi, Cipla, GlaxoSmithKline, Sanofi (to institution) and is Scientific advisor to AlbusHealth® and ProAxsis®. The rest of authors declare that they have no known competing financial interests or personal relationships that could have appeared to influence the work reported in this paper.

## Results

Each of the four pipelines resulted in similarity scores primarily driven by a unique set of features. It was demonstrated that data transformations according to each pipeline prior to clustering can result in a variation of clustering results of over 40%. The most appropriate pipeline was selected on the basis of feature ranking and clinical expertise. There was moderate agreement between clinicians as measured by Cohen's kappa coefficient.

## Conclusions

Data transformation has downstream and unforeseen consequences in cluster analysis. Rather than viewing this process as a black box, we have shown ways to quantitatively and qualitatively evaluate and select the appropriate preprocessing pipeline.

## Introduction

Diseases such as chronic obstructive pulmonary disease (COPD) are characterised by high heterogeneity. Phenotyping studies based on cluster analysis are increasingly employed in the effort to discover clinically informative subgroups of patients in order to better tailor treatment. For all clustering algorithms, a measure of similarity (or dissimilarity) between data points is the basis on which clustering algorithms partition the dataset. This crucial input depends on the choice of features and their relative weights, and as such "can only come from subject matter considerations [1]."

Calculating similarity between cases is also required for certain classes of supervised learning algorithms used in prediction studies [2], most notably for K nearest neighbours, as well as in the application of decision support techniques such as case-based reasoning [3].

### Patient representation

In machine learning "representation learning" is the process of applying unsupervised techniques on a dataset, in order to arrive at a more useful set of features [4]. The methods used to construct the new representation can vary from simple data scaling and linear algebra transformations to deep neural networks. The aims of constructing a representation are many-fold and include preserving the feature relationships of the original representation in a lower dimensional space, obtaining uncorrelated features, and noise reduction.

The potential number of features increases dramatically, especially when electronic health records are used, while datasets can be linked together, driving the number of features even higher, often in the hundreds or thousands. Several difficulties arise when working in high-dimensional spaces. With each additional feature, the volume of the space represented grows so quickly that the data cannot keep up and thus becomes sparse. As the dimensionality increases, the data fills less and less of the data space. As a result, in order to maintain an accurate representation of the space, the size of the input vector needs to grow exponentially [5]. This is sometimes referred to as "the curse of dimensionality" [6].

Another aspect of the curse of dimensionality that directly impacts on the notion of data-point similarity is the tendency of pairwise distances in high dimensions to become almost equal [7], therefore rendering techniques such as k-means and k-nearest neighbours meaningless. Representation learning therefore virtually always involves data transformation methods that aim to achieve some degree of dimensionality reduction.

Datasets with mixed-type features pose an additional challenge to similarity calculation for the purpose of cluster analysis. Similarity can only be calculated on one data type at a time, and combining similarity of different data types is a non-trivial task [8]. Many different practices are used to transform mixed data into a single type. These can include the practice of binning numerical data, treating ordinal data as continuous and turning categorical features into binary ones (one-hot encoding [9]), also referred to as "dummy variables". All of these practices come at a cost, as feature manipulation combined with dataset transformations can result in unexpected feature "dominance" effects that can result from unintended weighting of certain data types. This becomes important downstream, when cluster algorithms are used to partition the dataset.

## Study aims

The primary aim of this work was to describe and implement an approach to evaluating data representations and subsequent impact on data-point similarity resulting from a variety of data processing pipelines. This evaluation includes:

1. The investigation of assigned feature importance in calculating data point similarity, including the relative contributions of numeric and categorical features

2. The clinical evaluation of resulting similarity relationships by expert clinician raters

3. The evaluation of cluster tendency of the resulting representations

These three evaluation elements can be individually considered in order to select an appropriate processing algorithm for the desired application, in this case, cluster analysis, hence the inclusion of cluster tendency as an additional metric.

The secondary aim of this work was to demonstrate that decisions on data pre-processing have downstream effects on clustering results.

## Methods

### Literature survey

In order to gain an overview of the common practices with regards to mixed data handling in subtyping for COPD, we examined three systematic review [10–12] and expanded the search (non-systematic) to further studies from the most recent literature. We also discuss the findings of a relevant systematic review by Horne et al. [13], which focused on the methodological challenges of cluster analysis using mixed-type data in asthma.

### Data sources

The dataset for this study consisted of anonymised, routinely collected healthcare data provided by the Clinical Practice Research Datalink. CPRD provides anthropometric measurements, laboratory tests, clinical diagnoses, symptoms, prescriptions, and medical procedures, coded with the Read controlled clinical terminology. The primary care practices in CPRD and the subset of linked practices used in the present analysis are representative of the UK primary care setting [14] and have been validated for epidemiological research [15].

Linked mortality data from the Office for National Statistics (ONS) was provided for this study by CPRD. Data is linked by NHS Digital, the statutory trusted third party for linking data, using identifiable data held only by NHS Digital.

We selected anonymized patient EHR using code lists derived from the CALIBER © resource (https://www.ucl.ac.uk/health-informatics/caliber and https://www.caliberresearch.

org/). CALIBER, led from the UCL Institute of Health Informatics, is a research resource providing validated electronic health record phenotyping algorithms and tools for national structured data sources [16–18].

## Ethics approval and consent to participate

In all CPRD studies, consent is not given on an individual patient level. Selected general practices consent to this process at a practice level, with individual patients having the right to opt-out. A protocol for this research was approved by the Independent Scientific Advisory Committee (ISAC) for MHRA Database Research (protocol number 16_152Mn). Generic ethical approval for observational research using CPRD with approval from ISAC has been granted by a Health Research Authority (HRA) Research Ethics Committee (East Midlands–Derby, REC reference number 05/MRE04/87).

## Population definition

We used validated algorithms and robust phenotyping approaches which have been evaluated and published previously. COPD diagnosis was based on a validated algorithm (86.5% PPV) used in over 50 publications [19] combined with either a current or former smoking status.

The study period was January 1st 1998 to January 3rd 2016, and individuals were eligible for inclusion if: a) they were (or turned) 35 years of age or older during the study period, b) they had been registered for at least one year in a primary care practice which met research data recording standards (known as Up To Standard and defined using CPRD algorithms examining patterns of data completeness and temporal gaps in recording) and c) had at least one diagnostic code for COPD. We did not impose an upper age limit. We used an open cohort design, so patients entered the study when they met the inclusion criteria. We set the index date for each participant to the date of the first COPD diagnosis code recorded in primary care while the participant was eligible. Study exit date was defined as the earliest of Last Collection Date, Date the patient exited the practice, date of death or end of the study period. Patients with any missing data were excluded from the analysis.

## Feature selection

The following features were selected as input to the analysis on the basis of their clinical relevance to COPD as described previously [20]: sex, body mass index (BMI), smoking status (current or ex), personal history of atopy, airflow obstruction as defined by the global initiative for chronic obstructive lung disease (GOLD) stage [21]: 1 (FEV1% predicted $> = 80\%$), 2 ($50\% < = $ FEV1%predicted $< 80\%$), 3 ($30\% < = $ FEV1% predicted $< 50\%$) and 4($< = $ FEV1% predicted $< 30\%$), eosinophil % of white blood cell count, gastro-esophageal reflux disease (GERD), chronic rhinosinusitis (CRS), diabetes, anxiety, depression, ischaemic heart disease (IHD), hypertension, and heart failure.

We classified COPD therapy type with regards to different combinations of inhaled corticosteroids (ICS), Long Acting Muscarinic Antagonists (LAMA) and/or Long Acting Beta-2 Antagonists (LABA) as: a) no therapy (none of LAMA, LABA prescribed), b) mono-therapy (prescription of LABA or LAMA only), c) dual therapy (prescription of either LABA&LAMA or LABA&ICS or LAMA&ICS), and c) triple therapy: prescription of all LABA, LAMA and ICS.

Phenotyping algorithms for all features were defined in CALIBER using previously-published phenotypes which have been used in over 60 publications [18, 22–25].

## Data processing pipelines

The data processing pipelines implemented include a combination of: data scaling, principal component analysis (PCA) [26], multiple correspondence analysis (MCA) [27], and an auto-encoder network [28]. The dataset was randomly split into a training and test set in order to evaluate the generalisation of each pipeline on unseen data.

## Principal component analysis

The aim of PCA is to reduce the dimensionality of a data set consisting of a large number of interrelated features, while retaining as much as possible of the variation present in the data set. This is achieved by transforming to a new set of features, the principal components, which are uncorrelated, and which are ordered so that the first few retain most of the variation present in all of the original features [26]. PCA was used on the numerical features of each dataset.

## Multiple correspondence analysis

MCA is a method used to analyse the pattern of relationships of categorical variables. MCA also performs dimensionality reduction by representing data as points in a low-dimensional Euclidean space, projected onto orthogonal vectors. As such, MCA can be seen as analogous to PCA, which is used to reduce correlation between numerical features. When this method is used on mixed data-types datasets it requires binning of numerical features and treats ordinal data as categorical.

## Autoencoders

An autoencoder is a type of artificial neural network that is used to learn a lower dimensional representation (encoding) for a set of data [28]. The network encodes the data in the lower dimension and also "learns" a reconstruction algorithm that returns the original input. Auto-encoders operate under the heuristic that the most relevant features in the data capture the information necessary to linearly reconstruct the original data as accurately as possible. As such, the data is first encoded in a lower dimension, then trained to recover the original input from the encoding.

Autoencoders can be constructed to be simple or "shallow" or "deep", with multiple hidden layers. The main difference in autoencoders is that the loss function is calculated on the difference of the output from the original input rather than a response variable.

Unlike MCA or PCA, which when applied have a unique solution resulting from specific mathematical transformations, autoencoders acting on a dataset can result in different representations which depend on the hyperparameter combination and the number of training cycles. The main advantage of using autoencoders versus PCA and MCA is their ability to learn non-linear transformations. This ability comes at a computational cost, as autoencoders attempt to solve a non-linear optimisation problem [5].

The autoencoder network consists of the following types of layers: An input layer, a number of hidden layers, a bottleneck layer (which contains the learned representation vectors) and one output layer. A schematic architecture of an autoencoder neural network with a single and two hidden layers respectively is shown in Fig 1.

Autoencoder hyperparameter selection was performed using a grid search approach in which every possible combination of a set of parameters is explored, by training the algorithm on the training set and evaluating the results based on a loss function (mean squared error) calculated on both the training and test dataset. The selected algorithm was the one which delivered the best performance on the test set.

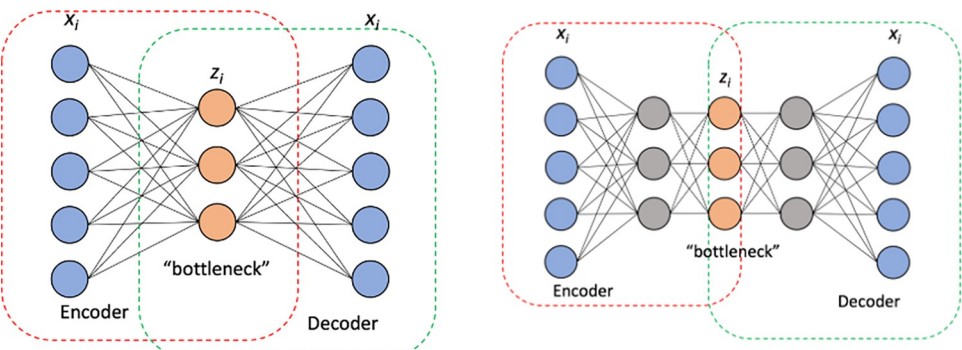

**Fig 1. Network architecture for the two types of AE used in the analysis.** (a) Single hidden layer and (b) Two-hidden layers. In both cases, $x_i$ represent elements of the input as well as the output vector. $z_i$ represents elements of the learned lower dimensional representation. (a) Shallow autoencoder architecture and (b) Deeper autoencoder architecture.

## Patient representations

Four different data processing pipelines were applied and evaluated on the training set:

1. Binning of numerical features and MCA on the full feature set (MCA, Fig 2),

2. MCA on categorical and PCA on numerical features (MCA/PCA, Fig 3),

3. MCA on categorical and PCA on numerical features, followed by PCA on the resulting two sets of components (MCA/PCA/PCA, Fig 4) and

4. Feature scaling followed by input into autoencoder neural network (AE, Fig 5).

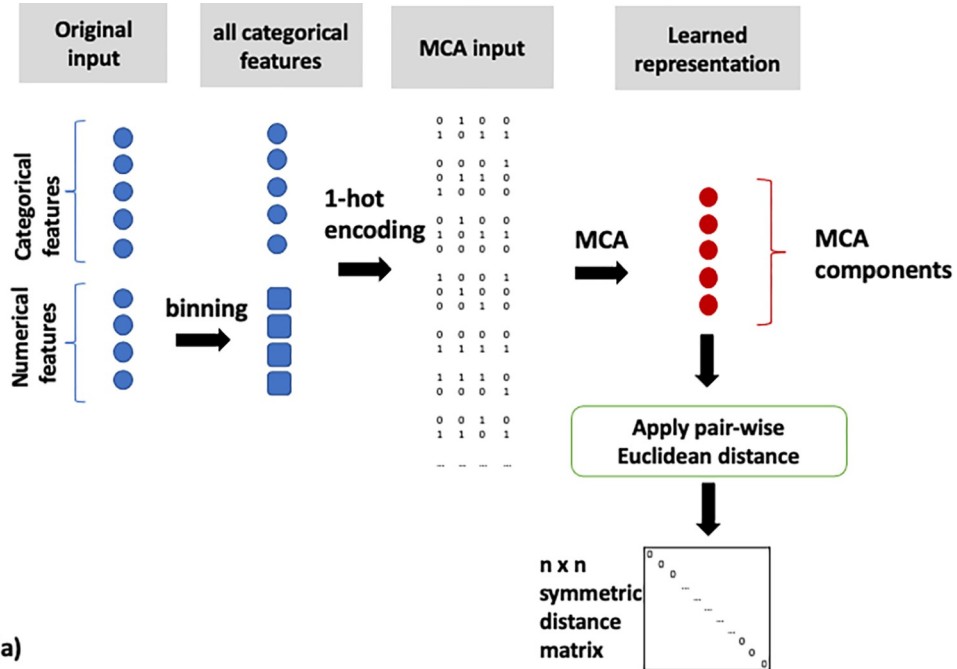

**Fig 2. Flow diagram of the MCA data analysis pipeline.** MCA on all features after numerical feature categorisation.

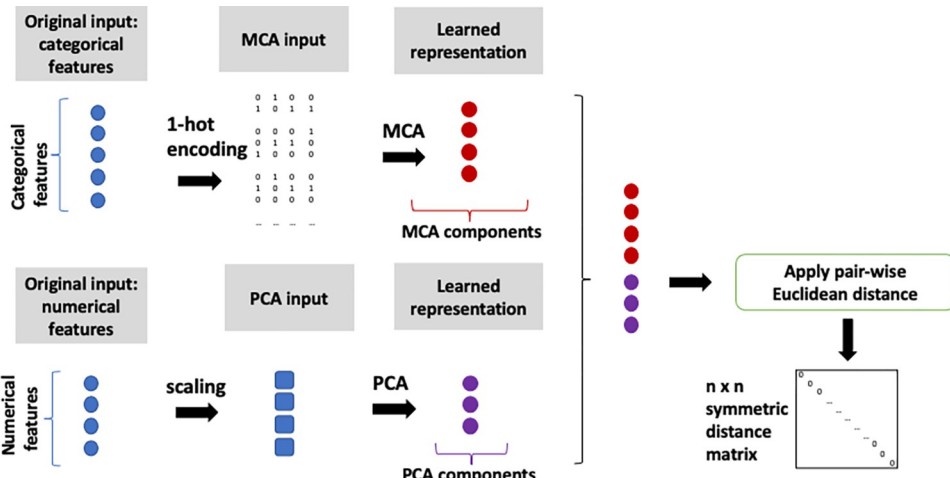

(b)

**Fig 3. Flow diagram of the MCA/PCA data analysis pipeline.** MCA on categorical and PCA on numerical features.

Pairwise similarity scores were computed for all patients on the basis of the representations derived from the four pipelines described above. The distance metric used to calculate these similarities was the Euclidean distance, with a lower distance indicating closer similarity. A schematic of the steps implemented for each pipeline is shown in Figs 2–5. All data transformations fitted on the training set were applied to the test set.

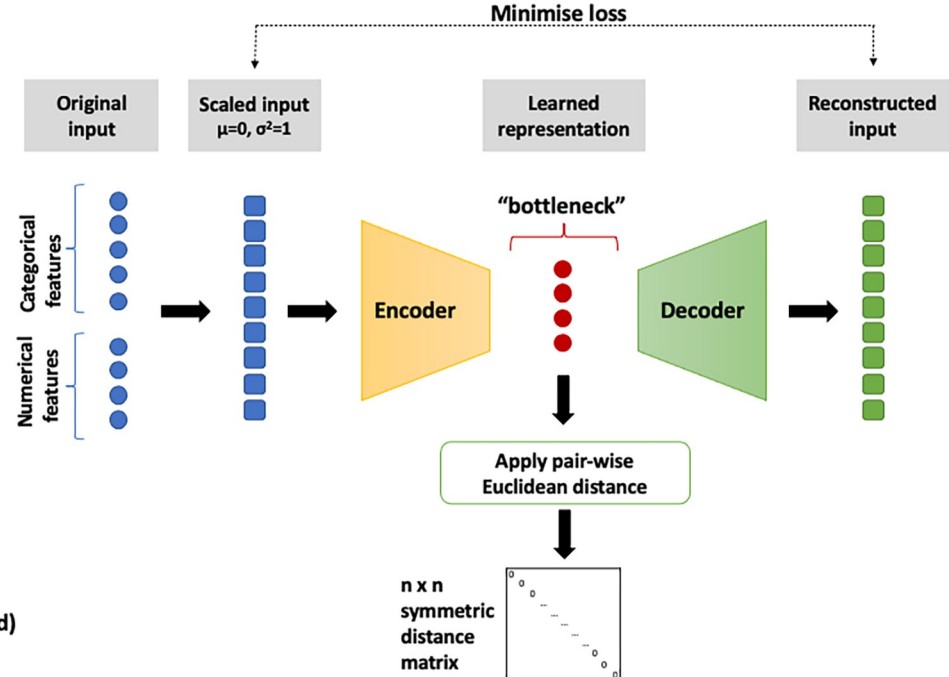

(d)

**Fig 4. Flow diagram of the MCA/PCA/PCA data analysis pipeline.** MCA on categorical and PCA on numerical features followed by PCA on resulting components of both methods.

The process above resulted in four similarity matrices, each containing the pairwise distances between all patients in the cohort. Given the four matrices derived for the same cohort, we now want to ask the question: To what extent does each feature contribute towards the similarity matrix? To answer this question we have devised a metric, described in the following section.

### Feature importance for patient similarity

In order to quantify the contribution of each feature to the resulting patient similarity, we devised a new metric, called relative variability (RV). RV is a ratio comparing feature variability within the closest N neighbours (according to the distance matrix) of each observation to the feature variability in the overall cohort. The way feature variability is captured differs depending on the feature data type. The RV is then averaged for each feature over all patients in the cohort. A detailed explanation is given below:

1. Pairwise numerical and ordinal feature differences were calculated between all patients in the cohort as follows:

$$d_x^{\ ij} = |x_i - x_j|$$

where $x_i$, $x_j$ the values of feature $x$ for patients $i$ and $j$ respectively.

2. A measure of the average absolute feature difference between each pair of patients $q_x^{cohort}$ was calculated for numerical features and ordinal features on the basis of the median and mean difference $d_x^{ij}$ respectively:

3. $q_x^{cohort} = median(d_x^{\ ij}) \ \forall \ i = 1(1)m, j = 1(1)m, i \neq j$ Numerical features
   $q_x^{cohort} = mean(d_x^{\ ij}) \ \forall \ i = 1(1)m, j = 1(1)m, i \neq j$ Ordinal features
   where $m$ is the number of patients in the cohort.

4. For categorical features, as a distance metric, we calculated the proportion of disagreement between each pair of patients in the cohort

$$p_x^{cohort} = number \ of \ pairs \ in \ disagreement / total \ number \ of \ pairs$$

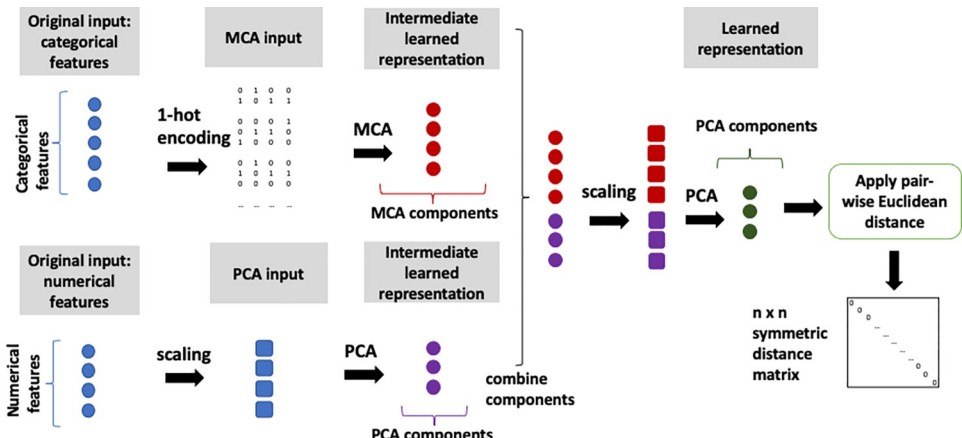

**(c)**

**Fig 5. Flow diagram of the AE data analysis pipeline.** Feature scaling followed by application multiple layer autoencoder neural network.

5. $q_x^{cohort}$ and $p_x^{cohort}$ were used as measures of each feature variability within the overall cohort.

6. We selected a sample of $N$ closest neighbours for every patient, and for each feature calculated the average absolute difference between this patient and their $N$ neighbours "$q_{sample}$" and proportion of disagreement "$p_{sample}$", as per the previous steps. $q_{sample, N}$ and "$p_{sample, N}$" were therefore used as a measure of feature variability in the immediate neighbourhood of the patient, as represented in lower dimensional space.

7. Relative variability (RV) is defined as the ratios $q_{sample,N}/q_{cohort}$ x 100 and $p_{sample,N}/p_{cohort}$ x 100 and used it to calculate feature importance

8. Different $N$s were used in order to test the robustness of this approach. (results provided in supplementary data document)

RV Numbers close to 0 indicate high importance of this feature in the low dimensional space (most of the variability is eliminated in closest neighbours), whereas numbers close or above 100 indicate that this feature has similar or greater spread compared to the overall cohort and is therefore not driving patient similarity.

For each pipeline, feature RV was ranked in ascending order, with top rankings indicating the most important features. In order to compare the efficacy of each pipeline in minimising the variability between features, we defined a summative metric, called mean relative variability (MRV) as the mean of all RV for numerical (including ordinal) and categorical features respectively. The MRV can be seen as a type of cost-function metric. The lower the MRV, the more similar patients are to each other with regard to all features.

In order to evaluate whether feature importance for each pipeline also applies to unseen data, Ranked RV and MRV were also calculated on the test set data representations (results in S2 Table in S1 File).

## Cluster tendency

We used the Hopkins index [29] as a measure of cluster tendency for each of the four representations. The Hopkins index takes values between 0 and 1, where values close to 1 denote highly clustered data, values close to 0.5 indicate randomly drawn data, and values close to 0 indicate a uniform distribution (regularly spaced data).

## Effect of representation on clustering results

Having obtained 4 low dimensional representations of the cohort, the next step was to investigate whether the differences in feature importance we described have a bearing on clustering results. We used the representation output of all four pipelines as input to the popular k-means clustering algorithm with values of k between 2 and 5, using identical random initialisations. We thus obtained 4 sets of clusters for each method.

We then compared cluster assignment congruity by calculating the percentage of patients that are assigned to the same cluster between pipelines. This allowed us to evaluate the practical implication of different representations on the output of the clustering algorithm. A high percentage of co-assignment indicates that irrespective of the pipeline used to obtain the input representation, the cluster composition does not change. Conversely, a low percentage of co-assignment indicates that pipeline choice makes a difference to the resulting clustering results.

## Clinical evaluation of patient similarity

A sample of 125 patients was selected at random as references. The best match for each reference patient was selected based on the pairwise matrices of each representation. For cases

where the best match was an identical patient (in terms of all features used in the analysis), the second or $n^{th}$ non-identical patient was selected (where n-1 is the number of identical patients to the reference patient). The choice of sample size was pragmatic, based on a pilot study of the time it took a clinician to complete 10 ratings, in order to complete the ratings in one sitting, avoid concentration fatigue and to limit the exercise to one hour in total.

The reference patient and case matches were presented to two clinical experts (Consultants in Respiratory medicine) in the form of clinical vignettes on a multiple-choice web tool (S1 Fig in S1 File). The tool allows the experts to blindly (i.e. not knowing which pipeline they are rating) and independently rank all matches.

In order to assess interrater agreement between the two clinical experts, we calculate raw percentage agreement scores as well as Cohen's kappa coefficient [30], which is used to estimate the proportion of agreement that is not attributed to chance. Confidence intervals were calculated as per Fleiss et al. [31].

## Results

### Literature survey: Handling mixed-data types in COPD subtyping studies

11 out of 24 studies identified involved mixed data types. Of those, one study dichotomised continuous features resulting in all binary feature analysis [32]. Two studies [33, 34] excluded features based on correlations/additive effects with no discussion of mixed-type issues. Two studies [35, 36] performed dimensionality reduction separately for categorical (MCA) and numerical (PCA) features, combining the resulting components for clustering. Another study [37] treated features separately also with MCA/PCA but also used multidimensional scaling on all features combined. One study [38] used an unsupervised random forest metric [39] specifically to deal with the mixed data problem. Finally, one study [20] binned numerical features in order to combine with categorical ones and used the MCA-derived components as input to the clustering algorithm. The processing of mixed-type data was not documented for two of the studies [40, 41].

Horne et al. [13] systematically reviewed publications with similar aims to our literature search on COPD, describing asthma subtyping studies. The methods commonly used for feature selection and feature transformation were MCA, PCA, and factor analysis (FA) [42], a technique similar to PCA. Overall: "Most studies explicitly stated the clustering method that they used but were less explicit regarding the preprocessing steps and choice of dissimilarity measure." [13]. The Euclidean distance metric was used on mixed-type data in over half the studies, without consideration of the potential, unintended, weighting implications of performing this calculation without the use of feature scaling or feature encoding.

### Cohort characteristics

The study comprised 30,961 patients from 393 primary care practices. The characteristics, which include all features used in the analysis of the overall cohort, and the training and testing dataset partition are shown in Table 1.

### Representation approaches using MCA and PCA

The number of components retained as the learned representation was determined by requiring a threshold of 75% of the variance. The rationale for this approach is that each feature below the threshold is not adding any significant value to the percentage of explained variance and can therefore be discarded. This was the case for both MCA- and PCA-based approaches, leading to a total number of components as follows:

- MCA:          3 components

- MCA/PCA      7 components

- MCA/PCA/PCA      6 components

## Optimal autoencoder architecture

An exhaustive hyperparameter search was run in order to select the best autoencoder with regards to number of layers (ranging in [2–5]), number of hidden units (ranging in [10, 12, 16]), and a learning rate (ranging in [0.1, 0.01, 0.001, 0.0001]).

The top performing autoencoder consisted of 3 deep layers of 16 hidden units each, a learning rate of 0.001 with a predetermined 6 unit bottleneck layer. A 6 unit bottleneck was selected to match the pipeline with the intermediate number of components (MCA/PCA/PCA).

## Feature importance

For each representation, a measure of feature importance was calculated using the RV metric on a sample size of 20 for each feature, and the results were ranked in Table 2. The way to interpret this metric is as follows: If we take the example of the MCA top ranking feature "anxiety", an RV value of 39.2% signifies that if x pairs patients out of the total N have different status with regards to anxiety diagnosis, then according to the MCA representation, for the 20 closest neighbours of each patient, this number is 0.394x, meaning that they are more similar to each other than the overall cohort with regards to anxiety diagnostic status—which can either mean they are on the whole more or less likely to be diagnosed with anxiety.

**Table 1. Characteristics of the COPD cohort (overall, training and test sets) used for measuring patient similarity.**
Mean value and standard deviation are presented for continuous feature.

|  | Entire cohort | Training set | Test set |
|---|---|---|---|
| N | 30467 | 22899 | 7568 |
| Age at index | 67.0 (10.8) | 67.0 (10.9) | 67.1 (10.8) |
| BMI (kg/m$^2$) | 27.5 (6.1) | 27.6 (6.1) | 27.5 (6.1) |
| FEV$_1$% predicted | 66.1 (20.9) | 66.1 (20.9) | 66.4 (21.1) |
| Eosinophils (% WBC) | 3.0 (1.8) | 3.0 (1.8) | 3.0 (1.8) |
| Sex (N, % male) | 16559 (54.5) | 12391 (54.1) | 4168 (55.1) |
| Smoking (N, % current) | 16299 (53.5) | 12239 (53.4) | 4060 (53.6) |
| Anxiety (N, %) | 3075 (10.1) | 3713 (10.0) | 787 (10.4) |
| Depression (N, %) | 3370 (11.1) | 2560 (11.2) | 810 (10.7) |
| Atopy (N, %) | 3715 (12.2) | 2774 (12.1) | 941 (12.4) |
| CRS (N, %) | 569 (1.9) | 424 (1.9) | 145 (1.9) |
| Diabetes (N, %) | 4938 (16.2) | 3713 (16.2) | 1225 (16.2) |
| Hypertension (N, %) | 10364 (34) | 7800 (34.1) | 2564 (33.9) |
| Heart failure (N, %) | 4596 (15.1) | 3433 (15.0) | 1163 (15.4) |
| Ischaemic heart disease (N, %) | 7013 (23.0) | 5261 (23.0) | 1752 (23.2) |
| GERD (N, %) | 2726 (8.9) | 2039 (8.9) | 687 (9.1) |
| Therapy (N, %) |  |  |  |
| None | 11459 (37.6) | 8616 (37.6) | 2843 (37.6) |
| Mono | 4027 (13.2) | 3013 (13.2) | 1014 (13.4) |
| Dual | 10058 (33.0) | 7579 (33.1) | 2479 (32.8) |
| Triple | 4923 (16.2) | 3691 (16.1) | 1232 (16.3) |

If we consider the top 5 features by RV for each pipeline, no single feature ranked highly for all four pipelines, and smoking was the only feature present in the top 5 of three out of four pipelines. Atopy, diabetes, depression, and therapy each ranked at the top 5 of two pipelines.

Looking at the MRV metric for categorical and ordinal features, the pipeline which primarily optimised for this data type was the autoencoder, while the pipeline optimising MRV for numerical features was MCA/PCA. Lowest and therefore most optimised average MRV for categorical and numerical data combined was achieved by the autoencoder, while the highest and therefore least optimal MRV was the one produced by MCA alone.

## Cluster tendency

The values of Hopkins index for each of the representations were as follows:

- MCA: 0.94

- MCA/PCA: 0.84

- MCA/PCA/PCA: 0.86

- Autoencoder: 0.98

According therefore to the Hopkins index, the Autoencoder representation had the highest cluster tendency, while the MCA/PCA representation had the lowest cluster tendency.

## Impact of representation on cluster analysis

Our next aim was to evaluate how a clustering algorithm applied on the same training dataset given in four different representation methods would group the same patients. We used the popular k-means algorithm with values of k between 2 and 5 and calculated the percentage of patients that are assigned to the same cluster between each representation (Table 3).

We can see that irrespective of k, up to more than 40% of patients do not cluster together, despite coming from the exact same original dataset with the same number of features.

In order to evaluate the sensitivity of the results presented in Table 3, we performed 20 repeats of clustering using subsets (bootstraps) of 10% of the data for each representation. We evaluated the percentage of patient pairs that were allocated to the same cluster in each iteration. The results are presented in the diagonal of Table 3.

## Clinical evaluation of patient similarity

Expert clinicians ranked the most similar patients proposed by each pipeline between 1 (best proposal) and 4 (worst proposal). Assigning the same ranking to more than one pipeline was allowed, as long as a 1 and 4 were assigned to at least one pipeline each. The ranking results are summarised in Fig 6.

Rater 1 assigned best ranking to the MCA/PCA pipeline, followed by MCA/PCA/PCA, while rater 2 assigned best to the MCA/PCA/PCA pipeline followed by the Autoencoder. On average, the most highly ranked pipeline by both raters was MCA/PCA/PCA, having an average best assignment score of 56.4%, more than half of the ratings. This was followed by MCA/PCA at 45.6% and autoencoder at 39.2%.

Both raters ranked MCA as the worst-scoring pipeline in the majority of the ratings.

## Summary of evaluation results

We used a variety of metrics to evaluate the four data representations. Table 4 below summarises the results. The selection of relevant metrics will ultimately depend on the use case. For

**Table 2. Features ranked by degree of influence on the resulting similarity according to each pipeline.** Bolded entries indicate numerical features.

| Rank | MCA (% RV) | MCA/PCA (% RV) | MCA/PCA/PCA (% RV) | AE (% RV) |
|---|---|---|---|---|
| 1 | Anxiety (39.2) | **FEV$_1$% pred (28.2)** | CRS (0.1) | Atopy (0.1) |
| 2 | Depression (41.0) | **Age at index (28.9)** | Therapy (5.4) | IHD (1.9) |
| 3 | Diabetes (45.8) | **BMI (31.6)** | Atopy (5.5) | Sex (4.9) |
| 4 | Smoking (53.4) | **Eosinophils (32.7)** | Smoking (30.4) | Therapy (4.9) |
| 5 | Heart failure (55.2) | Depression (35.5) | Diabetes (33.0) | Smoking (5.4) |
| 6 | IHD (59.7) | Anxiety (37.2) | Depression (36.0) | GERD (8.5) |
| 7 | Sex (63.5) | Smoking (40.5) | **FEV$_1$% pred (36.5)** | Diabetes (8.8) |
| 8 | **Age (64.7)** | IHD (45.2) | Anxiety (36.9) | Heart failure (12.8) |
| 9 | **BMI (66.4)** | Heart failure (48.7) | **Age (37.1)** | Hypertension (19.5) |
| 10 | Hypertension (79.7) | Sex (49.7) | IHD (38.3) | Anxiety (21.2) |
| 11 | Atopy (89.1) | CRS (52.2) | **BMI (40.9)** | Depression (32.1) |
| 12 | GERD (89.1) | Atopy (61.5) | **Eosinophils (41.8)** | **BMI (62.6)** |
| 13 | **FEV1% pred (91.4)** | Therapy (64.3) | Heart failure (43.6) | CRS (68.5) |
| 14 | CRS (95.5) | Diabetes (72.2) | Sex (48.4) | **FEV$_1$% pred (69.4)** |
| 15 | Therapy (99.3) | GERD (72.5) | Hypertension (51.9) | **Age (73.6)** |
| 16 | **Eosinophils (99.5)** | Hypertension (88.1) | GERD (53.3) | **Eosinophils (93.8)** |
| **MRV categorical** | 67.6 | 55.6 | 27.3 | 15.7 |
| **MRV numeric** | 80.5 | 30.4 | 39.1 | 74.9 |
| **MVR overall** | 70.8 | 40.3 | 33.7 | 30.5 |

example, while it might be desirable to compute feature importance for representations fed to a supervised learning model, the Hopkins index is only relevant to cluster analysis.

## Interrater agreement

We used Cohen's kappa coefficient to measure inter-rater agreement in the ranking of patient similarity. Given that each clinician was able to assign a ranking between 1 and 4, percentage agreement and Cohen's kappa become difficult to interpret, especially as Cohen's kappa thresholds for goodness of agreement primarily are more intuitive in the case of binary labels [43, 44].

We, therefore, decided to include a more interpretable metric of agreement by simplifying the agreement labels to "is the pipeline ranked the best" and "is the pipeline ranked the worst" respectively. The results of overall agreement on the basis of the original ranking as well as the binary (best/worst) metrics are summarised in Table 5.

**Table 3. Clustering results comparison for COPD cohort—different k values presented above and below diagonal as indicated by backslash "\".** The values of the diagonals represent averaged results obtained from 10% bootstrapped sampling and re-clustering.

| k = 2 \ k = 3 | MCA | MCA/PCA | MCA/PCA/PCA | AE |
|---|---|---|---|---|
| **MCA** | 98.2% \ 97.4% | 69.1% | 63.2% | 60.6% |
| **MCA/PCA** | 81.8% | 92.4% \ 93.3% | 69.9% | 57.7% |
| **MCA/PCA/PCA** | 80.4% | 92.5% | 92.0% \ 85.5% | 58.9% |
| **AE** | 61.1% | 60.4% | 61.3% | 99.9% \ 98.9% |
| **k = 4 \ k = 5** | **MCA** | **MCA/PCA** | **MCA/PCA/PCA** | **AE** |
| **MCA** | 95.0% \ 88.9% | 72.5% | 61.8% | 66.1% |
| **MCA/PCA** | 70.1% | 92.1% \ 89.9% | 64.1% | 64.7% |
| **MCA/PCA/PCA** | 59.1% | 59.3% | 99.2% \ 95.0% | 58.0% |
| **AE** | 64.2% | 61.2% | 56.2% | 99.9% \ 98.9% |

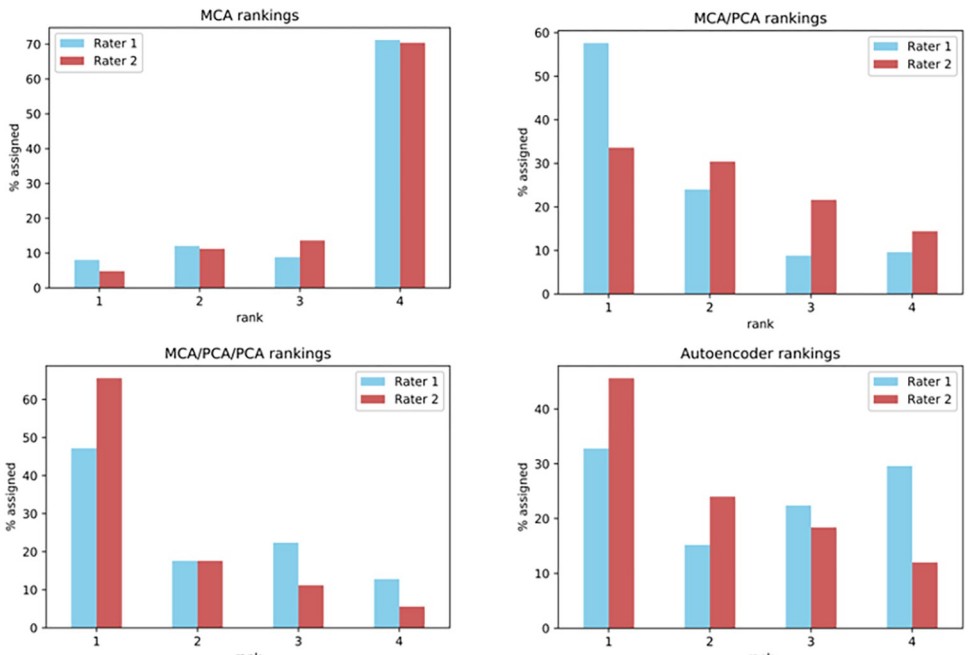

**Fig 6. Patient similarity rankings as assigned by two clinician raters for each pipeline.**

With kappa values ranging between 0.02 and 0.34, the two raters are far from perfect agreement, highlighting the subjective nature of deciding on "similarity" between patients. Nevertheless, in most cases there is a degree of non-chance agreement. Using the criteria set up by Landis and Koch [44], where a kappa above 0 and below 0.20 indicates slight agreement and kappa between 0.20–0.40 indicates fair agreement, there is agreement in both the overall and the best/worst kappas, with the exception of "best pipeline" for MCA and "worst pipeline" for MCA/PCA and MCA/PCA/PCA where the confidence intervals contain 0. It is, however, important to note that these are arbitrary thresholds and that there is no set threshold in the literature as to the interpretation of kappa magnitude.

## Discussion

In this work we have shown the importance of data transformation and dimensionality reduction methods in constructing patient similarity scores in particular when dealing with mixed data types. We have identified this area in the literature as lacking, and we saw in the brief literature survey that none of the studies discussed potential issues arising from the ways mixed-data types were handled.

Although all methods of data processing ultimately lead to a dimensionally reduced representation, their objectives and way of handling numerical and categorical features are different

**Table 4. Summary of evaluation results, including importance of features, cluster tendency and clinical expert evaluation for all four data processing pipelines.**

| | Feature Importance | | Cluster tendency | Clinical evaluation |
|---|---|---|---|---|
| Pipeline | Total MVR % | Categorical vs numeric | Hopkins Index | Average % "best" rating |
| MCA | 70.8 | balanced | 0.94 | 6.4 |
| MCA/PCA | 40.3 | favours numeric | 0.84 | 45.6 |
| MCA/PCA/PCA | 33.7 | most balanced | 0.86 | 56.4 |
| Autoencoder | 30.5 | favours categorical | 0.98 | 39.2 |

**Table 5. Rater agreement metrics.** Raw (%) agreement and kappa coefficient calculated on the basis of 1–2 as well as binary (best/worst) rankings.

| | MCA [95% CI] | MCA/PCA [95% CI] | MCA/PCA/PCA [95% CI] | AE [95% CI] |
|---|---|---|---|---|
| **Overall % agreement** | 0.62 [0.52, 0.7] | 0.39 [0.31, 0.48] | 0.46 [0.38, 0.56] | 0.44 [0.35, 0.53] |
| **% Agreement is best** | 0.14 [0.03, 0.44] | 0.39 [0.29, 0.5] | 0.48 [0.38, 0.59] | 0.42 [0.3,0.55] |
| **% Agreement is not best** | 0.9 [0.83, 0.95] | 0.46 [0.36, 0.57] | 0.26 [0.28, 0.5] | 0.58 [0.48, 0.68] |
| **% Agreement is worst** | 0.65 [0.56, 0.74] | 0.07 [0.03, 0.25] | 0.1 [0.02, 0.32] | 0.24 [0.13, 0.4] |
| **% Agreement is not worst** | 0.33 [0.21, 0.47] | 0.79 [0.7, 0.85] | 0.85 [0.77, 0.9] | 0.72 [0.63, 0.8] |
| **Overall kappa coefficient** | 0.18 [0.05, 0.31] | 0.13 [0.03, 0.24] | 0.15 [0.03, 0.26] | 0.24 [0.13, 0.35] |
| **Kappa "is best pipeline?"** | 0.20 [0, 0.49] | 0.24 [0.09, 0.39] | 0.23 [0.07, 0.39] | 0.34 [0.18, 0.5] |
| **Kappa "is worst pipeline"** | 0.28 [0.09, 0.48] | 0.02 [0, 0.2] | 0.1 [0, 0.32] | 0.26 [0.08, 0.43] |

from one another. PCA and MCA seek to uncorrelate the dataset and maximally explain the variance, while MCA can only be applied to binned numerical features, thus introducing an extra processing step that can influence its output and relative influence of categorical and numerical features. The extra PCA step in the MCA/PCA/PCA pipeline ensures that categorical and numerical feature eigenvectors undergo an additional transformation which balances their contributions.

The objective of an autoencoder is to "learn" the parameters of a forward and backward feature transformation such that applying the backward transformation will yield the closest result to the original features. Autoencoders can learn non-linear transformations and therefore produce a different representation compared to those of linear methods such as MCA and PCA.

By devising the relative variability metric we have quantitatively characterised the resulting representations with regards to feature importance in a way that is meaningfully comparable across different approaches of constructing representations. We have shown that feature importance is decided to a certain extent before the clustering algorithm is applied, during the data pre-processing and representation building step. We have also demonstrated the use of a qualitative approach to utilising clinical expertise in order to evaluate patient representations and in order to select the most appropriate pipeline for the given task.

Looking at all metrics—expert preference, cluster tendency, overall MRV, and balancing the MRV of categorical and numerical features- the preferred dimensionality reduction pipeline for this particular cohort is the MCA/PCA/PCA approach. It is perhaps not surprising that the algorithm preferred by the clinical experts was the one that most evenly balanced the contribution of numeric and categorical features.

It would be reasonable to challenge our decision to involve clinical experts in the evaluation of patient similarity. After all, unsupervised learning methods are used for data exploration and hypothesis generation, it would therefore make sense to eliminate all pre-existing bias. To this challenge we propose the following two responses:

1. As we have shown in this work, bias (towards certain features of the data) is inherent in the methods that we choose, and we are most often unaware of it. There is no reason to presuppose that this often-unmeasured bias is any different or better than clinical bias

2. When attempting to partition a large heterogeneous and complex dataset, it makes sense to sanity-check the similarity metrics. We have leveraged clinical expertise in order to exclude methods that suboptimally utilise the information contained in the dataset, while simultaneously taking into account quantitative information on each approach.

Nevertheless, the quantitative approach can be used in and of its own, as a simple heuristic to allow researchers an understanding of how the different methods handle the data, and place feature importance without any clinical input.

Finally, we have shown the impact of data transformations on the behaviour of cluster algorithms, which stems from the inherent absolute reliance of these algorithms to the pairwise distances between each data point, regardless of the method by which clustering itself takes place.

## Limitations

1. Representing patients in a lower dimensional space may limit the explainability of results, while the increase in data availability can introduce irrelevant features which can add noise to the calculated similarity metrics.

2. Effective measures of feature variability can be difficult to select, in particular with regard to ordinal features. Blair and Lacy [45] give a useful overview of potentially suitable metrics.

3. Comparison of feature variability between different feature data types is not particularly meaningful, however, mean relative variability can still be used as a cost-function metric.

4. The expert ranking exercise is difficult to implement when more than 20 or so features are used, and in that case, the quantitative metrics (RV, MRV) are more appropriate pipeline selection tools.

5. The comparative results obtained with each pipeline are dataset specific and would not generalise to any application. This is part of the motivation behind this work—which is to formalise the evaluation of data-processing pipelines as applied to a particular dataset.

## Conclusions

Even the simplest data transformation methods are often viewed as a black box, with no discussion on the magnitude of their impact to the downstream analysis. We have proposed a set of heuristics for analysing the effect of and selecting data processing pipelines. It is feasible and of potential interest to apply this analysis to more advanced, deep-learning approaches.

There is much scope to expand on the approaches to select the appropriate ways in which to make use of the ever-increasing breadth of data available. We, therefore, hope that this work will serve as a starting point for better evaluation of feature selection approaches and similarity definition in future studies.

## Supporting information

**S1 File.**
(ZIP)

## Acknowledgments

This study is based in part on data from the Clinical Practice Research Datalink obtained under licence from the UK Medicines and Healthcare products Regulatory Agency. The data is provided by patients and collected by the NHS as part of their care and support. The interpretation and conclusions contained in this study are those of the author/s alone

## Author Contributions

**Conceptualization:** Maria Pikoula, Spiros Denaxas.

**Data curation:** Maria Pikoula.

**Formal analysis:** Maria Pikoula.

**Funding acquisition:** Maria Pikoula.

**Investigation:** Maria Pikoula.

**Methodology:** Maria Pikoula, Constantinos Kallis, Sephora Madjiheurem, Spiros Denaxas.

**Project administration:** Maria Pikoula.

**Software:** Maria Pikoula, Spiros Denaxas.

**Supervision:** Spiros Denaxas.

**Validation:** Maria Pikoula, Jennifer K. Quint, Mona Bafadhel.

**Visualization:** Maria Pikoula.

**Writing – original draft:** Maria Pikoula.

**Writing – review & editing:** Maria Pikoula, Constantinos Kallis, Sephora Madjiheurem, Jennifer K. Quint, Mona Bafadhel, Spiros Denaxas.

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
