## [Decision Letter · Decision Letter 0]

6 Feb 2023

PONE-D-22-27760Semi-supervised evaluation of data processing pipelines on real-world electronic health records data for the purpose of measuring patient similarity.PLOS ONE

Dear Dr. Pikoula,

Thank you for submitting your manuscript to PLOS ONE. After careful consideration, we feel that it has merit but does not fully meet PLOS ONE’s publication criteria as it currently stands. Therefore, we invite you to submit a revised version of the manuscript that addresses the points raised during the review process.

We look forward to receiving your revised manuscript.

Kind regards,

Nguyen Quoc Khanh Le

Academic Editor

PLOS ONE

Journal Requirements:

2. Please provide additional details regarding participant consent. In the ethics statement in the Methods and online submission information, please ensure that you have specified what type you obtained (for instance, written or verbal, and if verbal, how it was documented and witnessed). If your study included minors, state whether you obtained consent from parents or guardians. If the need for consent was waived by the ethics committee, please include this information

Reviewers' comments:

Reviewer's Responses to Questions

**Comments to the Author**

1. Is the manuscript technically sound, and do the data support the conclusions?

Reviewer #1: No

Reviewer #2: Partly

2. Has the statistical analysis been performed appropriately and rigorously? 

Reviewer #1: Yes

Reviewer #2: Yes

3. Have the authors made all data underlying the findings in their manuscript fully available?

Reviewer #1: Yes

Reviewer #2: No

4. Is the manuscript presented in an intelligible fashion and written in standard English?

Reviewer #1: Yes

Reviewer #2: Yes

5. Review Comments to the Author

Reviewer #1: The aim of this paper was to outline and implement an analytical framework to evaluate four different data processing methods for constructing patient representations from EHR data for measuring patient similarity. The authors also evaluated the influence of features on patient similarity and the effect of different data processing methods on Euclidean distance-based cluster analysis.

This manuscript could be improved by addressing the following major and minor concerns.

Major concerns:

1. The term 'semi-supervised evaluation' was emphasized in the title of the manuscript. However, it was unclear how the semi-supervised evaluation was carried out. Typically, in a semi-supervised method, there was a small proportion of samples with supervised labels, and a large proportion of samples without supervised labels. What were the labels in the current study? How many samples had labels? How were these labels used in the evaluation?

2. The proposed data processing pipelines used PCA, MCA, and autoencoders for patient representation, which are less used in the era of deep learning. Therefore, the value of this manuscript is greatly discounted. I suggest that much more advanced representation methods should be used or reproduced and further evaluated.

3. Related to the previous comment, the references cited in the manuscript were too old. Up to 20 out of the 33 references were published 10 years ago, even in the 1960s. The authors only reviewed literature dealing with mixed-data types in COPD subtyping studies. I don't think that's enough. My suggestion is to widen the scope of the search without limiting the diseases studied, because the proposed pipelines were not specific to COPD.

4. The term "learned representation" appeared many times throughout the text. As far as I know, PCA and MCA were not a kind of "learning" methods. Representations using either method were obtained only by computation, not by learning or training.

5. What is the relationship between the evaluation of patient representation methods and the identification of important features? Had these important features been clinically validated?

Minor comments

6. Table 1 is not referred to in the main text.

7. More indices, such as Hopkins statistics, Silhouette index, and Davies-Bouldin index, should be used to evaluate the clustering solutions.

8. Authors should provide more details of the two clinical experts, such as clinical profession and experience.

Reviewer #2: In their work, the authors investigate how data with different types of features (numeric, categorical, ordinal) can be processed to assess similarity between data instances without biasing similarity to the feature type. They do so within the setting of patients with COPD, and assess clinical agreement with the created patient clusters.

1. The introduction lacks a clearly stated objective, aim or hypothesis. The introduction seems to steer the reader towards an investigation of methods that can deal with a large feature space of mixed feature types in a bias-free manner. However, I don’t think this question can be confidently answered using the study setup presented in this work. The authors should edit the introduction to be more specific regarding the research question that their study attempts to answer.

2. The authors focus heavily on representing patients in a lower-dimensional space. This has two potential drawbacks:

a. Depending on the final aim of the data processing pipeline (e.g. supervised learning), this may limit explainability.

b. Not all features are equally valuable for assessing similarity between patients. In the dataset employed by the authors, all features are at least plausibly meaningful. However with more data becoming available, features may also insert noise, and create irrelevant dissimilarity.

I do not expect from the authors that they investigate other methods for this publication, but they might make note of such issues for further work.

3. Though the algorithms are in a sense agnostic to what features are clinically relevant for finding patients, the clinicians who performed the assessment likely do have their preference, i.e. for grouping smoking and non-smoking patients. Thus a feasible alternative to the representation-based methods presented by the authors for clustering similar patients is to use expert consensus on important clinical features and compute Gower’s distance between patients for sampling.

4. Which loss function was used to train auto-encoders?

5. Computing the relative variability metric requires some steps that are not properly explained:

a. The concept of pairwise agreement is used, but I did not understand how agreement is assessed.

b. A reference patient is required, but it is not clear how this patient is selected.

6. To what degree is the cluster analysis shown in 3.5 sensitive to the composition of the patient dataset? I.e. if the cluster analysis is repeated multiple times using the same method but with subsets (e.g. bootstraps) of the data, how often do patients cluster together in the same cluster? Currently it is unclear if the presented values in table 4 are due to inherent differences between representation methods, or are close to the upper limit of what may be expected given the dataset.

7. Please be advised that though the authors may be prohibited from sharing the raw data (even though they claim these are fully anonymised), PLOS ONE does require that the data underlying the presented results should be published, e.g. those underlying Figure 6. See https://journals.plos.org/plosone/s/materials-software-and-code-sharing for more information.

6. PLOS authors have the option to publish the peer review history of their article (what does this mean?). If published, this will include your full peer review and any attached files.

Reviewer #1: No

Reviewer #2: **Yes: **Alex Zwanenburg

---

## [Author Response · Author response to Decision Letter 0]

21 Mar 2023

Rebuttal and Revisions

“Evaluation of data processing pipelines on real-world electronic health records data for the purpose of measuring patient similarity”

Editor Comments:

Dear Dr. Pikoula,

Thank you for submitting your manuscript to PLOS ONE. After careful consideration, we feel that it has merit but does not fully meet PLOS ONE’s publication criteria as it currently stands. Therefore, we invite you to submit a revised version of the manuscript that addresses the points raised during the review process.

We look forward to receiving your revised manuscript.

Kind regards,

Nguyen Quoc Khanh Le

Academic Editor

PLOS ONE

Response to Editor:

Thank you for reviewing our manuscript and for the constructive feedback included in the review. We have performed all revisions as detailed below. All the relevant files have been included in the revised submission. Please note the change in the title of the article as per Reviewer #1 comment.

Journal Requirements:

We have revised the manuscript formatting style according to the guides.

2. Please provide additional details regarding participant consent. In the ethics statement in the Methods and online submission information, please ensure that you have specified what type you obtained (for instance, written or verbal, and if verbal, how it was documented and witnessed). If your study included minors, state whether you obtained consent from parents or guardians. If the need for consent was waived by the ethics committee, please include this information

We have provided further clarification on the data sources subsection and the ethics statement.

“In all studies using CPRD data, consent is not given on an individual patient level. Selected practices…”

Reviewers' comments:

Reviewer's Responses to Questions

Comments to the Author

1. Is the manuscript technically sound, and do the data support the conclusions?

Reviewer #1: No

Reviewer #2: Partly

2. Has the statistical analysis been performed appropriately and rigorously?

Reviewer #1: Yes

Reviewer #2: Yes

3. Have the authors made all data underlying the findings in their manuscript fully available?

Reviewer #1: Yes

Reviewer #2: No

4. Is the manuscript presented in an intelligible fashion and written in standard English?

Reviewer #1: Yes

Reviewer #2: Yes

5. Review Comments to the Author

Reviewer #1: The aim of this paper was to outline and implement an analytical framework to evaluate four different data processing methods for constructing patient representations from EHR data for measuring patient similarity. The authors also evaluated the influence of features on patient similarity and the effect of different data processing methods on Euclidean distance-based cluster analysis.

This manuscript could be improved by addressing the following major and minor concerns.

We thank Reviewer #1 for their comments and constructive feedback.

Major concerns:

1. The term 'semi-supervised evaluation' was emphasized in the title of the manuscript. However, it was unclear how the semi-supervised evaluation was carried out. Typically, in a semi-supervised method, there was a small proportion of samples with supervised labels, and a large proportion of samples without supervised labels. What were the labels in the current study? How many samples had labels? How were these labels used in the evaluation?

Thank you for raising this. The term “semi-supervised” has been used loosely to refer to the expert clinicians’ involvement in the selection of the best method, rather than the use of labels for a portion of the dataset. We propose a corrected title omitting the term “semi-supervised” and have corrected the use of the term where it appears in the main text: 

“Evaluation of data processing pipelines on real-world electronic health records data for the purpose of measuring patient similarity.”

2. The proposed data processing pipelines used PCA, MCA, and autoencoders for patient representation, which are less used in the era of deep learning. Therefore, the value of this manuscript is greatly discounted. I suggest that much more advanced representation methods should be used or reproduced and further evaluated.

Thank you for this comment. PCA and MCA are extremely popular and powerful methods that are being heavily utilized in the field of disease phenotyping, as evidenced by the results of our literature search. As a result, investigating their ability to represent complex data such as medical history information, is of interest and is valuable. We selected the autoencoder as a representative deep learning method. We agree that examining other more advanced representation methods would be interesting, but this would have to be in a new manuscript as it is outside the scope of our current research manuscript. 

We have revised the manuscript to include a sentence in the conclusion on page 17 accordingly:

“It is feasible and of potential interest to apply this analysis to more advanced, deep-learning approaches.”

3. Related to the previous comment, the references cited in the manuscript were too old. Up to 20 out of the 33 references were published 10 years ago, even in the 1960s. The authors only reviewed literature dealing with mixed-data types in COPD subtyping studies. I don't think that's enough. My suggestion is to widen the scope of the search without limiting the diseases studied, because the proposed pipelines were not specific to COPD.

There are three points to this comment that we have addressed as follows:

1. References as old as the 1960s: These are references pertaining to methodology, and we decided to select and cite early foundational references to these concepts, such as the “curse of dimensionality” (Belman, 1961) and statistical tools, such as Cohen’s kappa coefficient (Fleiss et al. 1969, Landis and Koch 1977) on the basis that they are in use currently as originally introduced.

2. Scope of the literature survey: We chose COPD as an example of a typical “umbrella term” disease entity that researchers have tried to sub-phenotype using cluster analysis. We, therefore, restricted the literature search to COPD papers as it was not our intention to systematically review every possible application of cluster analysis.

3. Inclusion of more recent studies in the literature survey: We recognise that our literature search can and should be updated. We have therefore added the following articles included in a recent review on COPD clustering, as well as a methodology-focused systematic review of dealing with mixed-type data in asthma clustering studies and updated the relevant resuts section “Literature survey: Handling Mixed-data types in COPD subtyping studies” on page 9: 

Burgel et al. A simple algorithm for the identification of clinical COPD phenotypes. Eur Resp Jl 2017

Yoon et al. Prediction of first acute exacerbation using COPD subtypes identified by cluster analysis. Int J Chron Obstruct Pulmon Dis, 2019

Nikolaou et al. COPD phenotypes and machine learning cluster analysis: A systematic review and feature research agenda. Respir Med, 2020

Horne et al. Challenges of Clustering Multimodal Clinical Data: Review of Applications in Asthma Subtyping. JMIR Med Inform, 2020

4. The term "learned representation" appeared many times throughout the text. As far as I know, PCA and MCA were not a kind of "learning" methods. Representations using either method were obtained only by computation, not by learning or training.

Thank you for pointing this out. We have revised the manuscript to ensure terminology is consistent with the methods used either simply as “representation” in the case of MCA/PCA methods or “learned representation” for the autoencoder.

5. What is the relationship between the evaluation of patient representation methods and the identification of important features? Had these important features been clinically validated?

The features used as input to the data processing pipelines have all been selected by respiratory consultant clinicians on the basis of their clinical relevance the diseases. We have included detailed information on feature selection and definitions in a new subheading under Methods “Feature selection” on page 4.

“Feature Selection

The following features were selected as input to the analysis on the basis of their clinical relevance to COPD as described previously(Pikoula et al. 2019): sex, body mass index (BMI), smoking status (current or ex), personal history of atopy, airflow obstruction as defined by the global initiative for chronic obstructive lung disease (GOLD) stage(Rabe et al. 2007): 1 (FEV1% predicted > = 80%), 2 (50% < = FEV1%predicted < 80%), 3 (30% < = FEV1% predicted < 50%) and 4(<= FEV1% predicted < 30%), eosinophil % of white blood cell count, gastro-esophageal reflux disease (GERD), chronic rhinosinusitis (CRS), diabetes, anxiety, depression, ischaemic heart disease (IHD), hypertension, and heart failure. 

We classified COPD therapy type with regards to different combinations of inhaled corticosteroids (ICS), Long Acting Muscarinic Antagonists (LAMA) and/or Long Acting Beta-2 Antagonists (LABA) as: a) no therapy (none of LAMA, LABA prescribed), b) mono-therapy (prescription of LABA or LAMA only), c) dual therapy (prescription of either LABA&LAMA or LABA&ICS or LAMA&ICS), and c) triple therapy: prescription of all LABA, LAMA and ICS.

Phenotyping algorithms for all features were defined in CALIBER using previously-published phenotypes which have been used in over 60 publications(Rapsomaniki et al. 2014; Daskalopoulou et al. 2016; Koudstaal et al. 2017; Gho et al. 2018; Morley et al. 2014).”

We calculate the relative variance metric as a measure of how much each of the original features contributes to the similarity matrix. This is a metric of “algorithmically assigned” feature importance rather than a metric of the clinical importance of the feature. 

In terms of method evaluation, the calculation of this metric allows the user to understand the handling of each feature and its relative importance in the construction of the similarity matrix. In practice, it can allow the user to spot when feature encoding results in the disproportionate and undesirable weighting of a certain type of feature, as can happen for example with categorical over numerical features. 

We have included a “summary of evaluation results” subsection and table (new Table 4) in the revised manuscript which summarises the evaluation of each pipeline according to feature importance, cluster tendency (as per comment 7, see below) and clinical expert ranking on page 14.

“Summary of evaluation results

We used a variety of metrics to evaluate the four data representations. Table 4 below summarises the results. The selection of relevant metrics will ultimately depend on the use case. For example, while it might be desirable to compute feature importance for representations fed to a supervised learning model, the Hopkins index is only relevant to cluster analysis.”

Minor comments

6. Table 1 is not referred to in the main text.

Thank you for pointing out this omission. We have added a new subsection under Results “Cohort characteristics” that references Table 1 on page 9.

“Cohort characteristics

The study comprised 30,961 patients from 393 primary care practices. The characteristics, which include all features used in the analysis of the overall cohort, and the training and testing dataset partition are shown in Table 1.”

7. More indices, such as Hopkins statistics, Silhouette index, and Davies-Bouldin index, should be used to evaluate the clustering solutions.

We used the percentage of patients that cluster together as an indicator of clustering similarity between each set of representations. The aim of this comparison was to illustrate the difference pre-processing of data can make to the clustering results. While the Silhouette and Davies-Bouldin indices are used to (internally) evaluate clustering solutions, we would suggest this is a separate research question to the one our study is aiming to answer, which is the choice of pre-processing the data prior to clustering.

The Hopkins index is however a valuable index to compute, as it is a measure of cluster tendency of the dataset, and can thus be applied to each representation. We thank the reviewer for this recommendation. We have calculated the Hopkins index and included it in our results and discussion as an additional potentially useful metric which can help guide the choice of representation intended for cluster analysis.

This can be found under the subheading Results: Cluster Tendency on page 12 as follows:

“The values of Hopkins index for each of the representations were as follows:

• MCA: 0.94

• MCA/PCA: 0.84

• MCA/PCA/PCA: 0.86

• Autoencoder: 0.98

According therefore to the Hopkins index, the Autoencoder representation had the highest cluster tendency, while the MCA/PCA representation had the lowest cluster tendency.”

8. Authors should provide more details of the two clinical experts, such as clinical profession and experience.

We have provided this information under Methods: Clinical evaluation of patient similarity on page 8:

“…The reference patient and case matches were presented to two clinical experts (Consultants in Respiratory medicine)…”

Reviewer #2: In their work, the authors investigate how data with different types of features (numeric, categorical, ordinal) can be processed to assess similarity between data instances without biasing similarity to the feature type. They do so within the setting of patients with COPD, and assess clinical agreement with the created patient clusters.

We thank Dr Zwanenburg for their comments and constructive feedback.

1. The introduction lacks a clearly stated objective, aim or hypothesis. The introduction seems to steer the reader towards an investigation of methods that can deal with a large feature space of mixed feature types in a bias-free manner. However, I don’t think this question can be confidently answered using the study setup presented in this work. The authors should edit the introduction to be more specific regarding the research question that their study attempts to answer.

We thank the reviewer for this important observation. We have included a “study aims” subsection under “Introduction” on page 3:

“Study aims

The primary aim of this work was to describe and implement an approach to evaluating data representations and subsequent impact on data-point similarity resulting from a variety of data processing pipelines. This evaluation includes: 

1. The investigation of assigned feature importance in calculating data point similarity, including the relative contributions of numeric and categorical features

2. The clinical evaluation of resulting similarity relationships by expert clinician raters

3. The evaluation of cluster tendency of the resulting representations

The above metrics can be individually considered in order to select an appropriate processing algorithm for the desired application, in this case cluster analysis, hence the inclusion of cluster tendency as an additional metric.

The secondary aim of this work was to demonstrate that decisions on data pre-processing have downstream effects on clustering results.”

2. The authors focus heavily on representing patients in a lower-dimensional space. This has two potential drawbacks:

a. Depending on the final aim of the data processing pipeline (e.g. supervised learning), this may limit explainability.

b. Not all features are equally valuable for assessing similarity between patients. In the dataset employed by the authors, all features are at least plausibly meaningful. However with more data becoming available, features may also insert noise, and create irrelevant dissimilarity.

I do not expect from the authors that they investigate other methods for this publication, but they might make note of such issues for further work.

We thank the reviewer for raising these important observations. We have added relevant sections under study limitations page 17:

“Representing patients in a lower dimensional space may limit the explainability of results, while the increase in data availability can introduce irrelevant features which can add noise to the calculated similarity metrics.”

3. Though the algorithms are in a sense agnostic to what features are clinically relevant for finding patients, the clinicians who performed the assessment likely do have their preference, i.e. for grouping smoking and non-smoking patients. Thus a feasible alternative to the representation-based methods presented by the authors for clustering similar patients is to use expert consensus on important clinical features and compute Gower’s distance between patients for sampling.

The reviewer makes an interesting point, however, we chose cluster analysis since it’s a widely used method. In clinical applications of cluster analysis, the selection of features is generally agreed upon by clinicians, as is the case in our study. 

We agree that a different metric could also be explored (and indeed although we use it in this study, our method is not limited to the Euclidean distance), however, the proposed Gower’s distance would only be appropriate for categorical features, and therefore the issue of handling mixed data types remains.

4. Which loss function was used to train auto-encoders?

The loss function used was the mean squared error. This has been updated in the text under the Methods: Autoencoders subheading on page 6: 

“…..the results based on a loss function (mean squared error) calculated on both the training and test dataset.”

5. Computing the relative variability metric requires some steps that are not properly explained:

a. The concept of pairwise agreement is used, but I did not understand how agreement is assessed.

b. A reference patient is required, but it is not clear how this patient is selected.

Thank you for this comment, we agree it is important to clarify these points in the text. 

a. We have corrected the text and specified pairwise disagreement, instead of agreement, as we are calculating a distance metric. Pairwise disagreement is a way to quantify dissimilarity in categorical features. Two data points either share the same value of a categorical feature, therefore are in agreement, or they have different values and are in disagreement. To quantify the average disagreement between a patient and a sample of their neighbouring patients for a categorical feature we used the proportion of disagreement.

b. We have removed the word “reference”, as it may confuse with the reference patient mentioned in the clinical evaluation section. Instead, we specify that the neighbouring-patient feature similarity is calculated for every patient.

6. To what degree is the cluster analysis shown in 3.5 sensitive to the composition of the patient dataset? I.e. if the cluster analysis is repeated multiple times using the same method but with subsets (e.g. bootstraps) of the data, how often do patients cluster together in the same cluster? Currently it is unclear if the presented values in table 4 are due to inherent differences between representation methods, or are close to the upper limit of what may be expected given the dataset.

Thank you for making this very important point and suggestion. We re-sampled 10% of the dataset 20 times for each method and each number of clusters k, and repeated the cluster analysis. We reported the average percentage of agreement, (percentage pairs of patients who remain clustered together in each sample) in the diagonals of Table 3.

7. Please be advised that though the authors may be prohibited from sharing the raw data (even though they claim these are fully anonymised), PLOS ONE does require that the data underlying the presented results should be published, e.g. those underlying Figure 6. See https://journals.plos.org/plosone/s/materials-software-and-code-sharing for more information.

We are unfortunately unable to share the raw data pertaining to the patient records used, and access is only possible for approved researchers, following CPRD’s Research Data Governance Process, as described in the following guide: https://cprd.com/safeguarding-patient-data ; this is in line with the data providers governance framework we operate under and while data are indeed anonymized, the risk of identifiability still remains. 

We have provided the raw ratings data underlying Figure 6. This is now included in the Supporting Information file.

---

## [Decision Letter · Decision Letter 1]

5 May 2023

PONE-D-22-27760R1Evaluation of data processing pipelines on real-world electronic health records data for the purpose of measuring patient similarity.PLOS ONE

Dear Dr. Pikoula,

Thank you for submitting your manuscript to PLOS ONE. After careful consideration, we feel that it has merit but does not fully meet PLOS ONE’s publication criteria as it currently stands. Therefore, we invite you to submit a revised version of the manuscript that addresses the points raised during the review process.

We look forward to receiving your revised manuscript.

Kind regards,

Nguyen Quoc Khanh Le

Academic Editor

PLOS ONE

Journal Requirements:

Reviewers' comments:

Reviewer's Responses to Questions

**Comments to the Author**

1. If the authors have adequately addressed your comments raised in a previous round of review and you feel that this manuscript is now acceptable for publication, you may indicate that here to bypass the “Comments to the Author” section, enter your conflict of interest statement in the “Confidential to Editor” section, and submit your "Accept" recommendation.

Reviewer #1: (No Response)

Reviewer #2: (No Response)

2. Is the manuscript technically sound, and do the data support the conclusions?

Reviewer #1: (No Response)

Reviewer #2: Yes

3. Has the statistical analysis been performed appropriately and rigorously? 

Reviewer #1: (No Response)

Reviewer #2: Yes

4. Have the authors made all data underlying the findings in their manuscript fully available?

Reviewer #1: (No Response)

Reviewer #2: Yes

5. Is the manuscript presented in an intelligible fashion and written in standard English?

Reviewer #1: (No Response)

Reviewer #2: Yes

6. Review Comments to the Author

Reviewer #1: (No Response)

Reviewer #2: I would like to thank the authors for addressing my previous concerns and questions. I have the following minor comment on the revised manuscript:

1. The study aims subsection contains a paragraph starting “The above metrics can be… “. It was not clear to me what metrics are being referred to.

7. PLOS authors have the option to publish the peer review history of their article (what does this mean?). If published, this will include your full peer review and any attached files.

Reviewer #1: No

Reviewer #2: **Yes: **Alex Zwanenburg

---

## [Author Response · Author response to Decision Letter 1]

6 May 2023

PONE-D-22-27760R1

Evaluation of data processing pipelines on real-world electronic health records data for the purpose of measuring patient similarity.

PLOS ONE

Dear Dr. Pikoula,

Thank you for submitting your manuscript to PLOS ONE. After careful consideration, we feel that it has merit but does not fully meet PLOS ONE’s publication criteria as it currently stands. Therefore, we invite you to submit a revised version of the manuscript that addresses the points raised during the review process.

We look forward to receiving your revised manuscript.

Kind regards,

Nguyen Quoc Khanh Le

Academic Editor

PLOS ONE

Response to Editor:

Thank you for reviewing our manuscript and for the constructive feedback included in the review. We have performed all revisions as detailed below. All the relevant files have been included in the revised submission. 

Journal Requirements:

We have reviewed all references and found no issues or retractions.

Reviewers' comments:

Reviewer's Responses to Questions

Comments to the Author

1. If the authors have adequately addressed your comments raised in a previous round of review and you feel that this manuscript is now acceptable for publication, you may indicate that here to bypass the “Comments to the Author” section, enter your conflict of interest statement in the “Confidential to Editor” section, and submit your "Accept" recommendation.

Reviewer #1: (No Response)

Reviewer #2: (No Response)

2. Is the manuscript technically sound, and do the data support the conclusions?

Reviewer #1: (No Response)

Reviewer #2: Yes

3. Has the statistical analysis been performed appropriately and rigorously?

Reviewer #1: (No Response)

Reviewer #2: Yes

4. Have the authors made all data underlying the findings in their manuscript fully available?

Reviewer #1: (No Response)

Reviewer #2: Yes

5. Is the manuscript presented in an intelligible fashion and written in standard English?

Reviewer #1: (No Response)

Reviewer #2: Yes

6. Review Comments to the Author

Reviewer #1: (No Response)

Reviewer #2: I would like to thank the authors for addressing my previous concerns and questions. I have the following minor comment on the revised manuscript:

1. The study aims subsection contains a paragraph starting “The above metrics can be… “. It was not clear to me what metrics are being referred to.

Many thanks to Dr Zwanenburg for this comment and for your constructive feedback overall. We have rephrased the sentence to make it clear it refers to the three points detailed above:

“These three evaluation elements can be individually considered …”

7. PLOS authors have the option to publish the peer review history of their article (what does this mean?). If published, this will include your full peer review and any attached files.

Do you want your identity to be public for this peer review? For information about this choice, including consent withdrawal, please see our Privacy Policy.

Reviewer #1: No

Reviewer #2: Yes: Alex Zwanenburg

---

## [Editor Report · Decision Letter 2]

2 Jun 2023

Evaluation of data processing pipelines on real-world electronic health records data for the purpose of measuring patient similarity.

PONE-D-22-27760R2

Dear Dr. Pikoula,

We’re pleased to inform you that your manuscript has been judged scientifically suitable for publication and will be formally accepted for publication once it meets all outstanding technical requirements.

Kind regards,

Nguyen Quoc Khanh Le

Academic Editor

PLOS ONE
---

## [Editor Report · Acceptance letter]

7 Jun 2023

PONE-D-22-27760R2 

Evaluation of data processing pipelines on real-world electronic health records data for the purpose of measuring patient similarity. 

Dear Dr. Pikoula:

I'm pleased to inform you that your manuscript has been deemed suitable for publication in PLOS ONE. Congratulations! Your manuscript is now with our production department. 

Kind regards, 

on behalf of

Dr. Nguyen Quoc Khanh Le 

Academic Editor

PLOS ONE